# Evaluation on Strength Properties of Lime–Slag Stabilized Loess as Pavement Base Material

**Liang Jia [1], Li Zhang [1], Jian Guo [1], Kai Yao [2],\* , Sin Mei Lim [2] , Bin Li [3] and Hui Xu [2,4]**

[1]  College of Civil Engineering, Lanzhou University of Technology, Lanzhou 730050, China
[2]  Department of Civil and Environmental Engineering, National University of Singapore, Singapore 117576, Singapore
[3]  School of Transportation, Wuhan University of Technology, Wuhan 430063, China
[4]  School of Civil Engineering and Architecture, Zhejiang Sci-Tech University, Hangzhou 310018, China
\*   Correspondence: yaokai@u.nus.edu; Tel.: +65-90592451

**Abstract:** This study aimed to investigate the feasibility of using lime–slag stabilized loess as base-course material by assessing its unconfined compressive strength (UCS). Loess stabilized with various mix ratios were compacted and cured to three, five, seven, and 28 days, respectively, for further strength tests. The effects of binder content, lime-to-slag (L/S) ratio, porosity, and curing time on the UCS of stabilized loess were addressed in detail. The test results show that UCS increases with the increase in binder content or curing time, and it gains strength rapidly within the first seven days of curing. At the same binder content, UCS decreases with the decrease in L/S ratio or porosity. Finally, the correlations of UCS with binder content, porosity, and curing time were derived, which exhibited reasonable correlation coefficients $R^2$ (from 0.86 to 0.97).

**Keywords:** unconfined compressive strength; lime; slag; loess; curing time

## 1. Introduction

Loess is mainly distributed in the upper and middle part of China's Yellow River and the total distribution area is about 630,000 km², which is around 6.3% of the total land area of China [1–4]. Although loess in its dry state shows high strength and small deformation, it exhibits large-scale deformation and deterioration of mechanical properties when contacting with water, and this phenomenon is known as collapsibility [4–7]. The collapsibility of loess may cause problems, such as landslides, soil erosion, ground cracking, and settlement. These hazards will pose great threats to people's lives and property safety, as well as urban development in the loess area [7]. Therefore, collapsibility is inevitable during rainy seasons when loess is permanently soaked in the water. Previous studies revealed that the intrinsic characteristics of the porous structure of loess and the composition of cemented material are the main causes of collapsibility [8–11]. Given this, stabilization of the loess is of great significance to mitigate the occurrence of geologic hazards [6,10,12–15].

Some previous research results indicated that the application of stabilizing agent (such as cement or lime) can significantly improve the mechanical properties of soft soils [16–19]. The addition of lime has been a popular solution for loess stabilization, due to low cost and high technical efficiency in engineering applications [7,12]. However, the early-stage strength of lime stabilized soil tends to be quite low due to the slow pozzolanic reactions [20–26]. Many of the aforementioned studies demonstrate that lime can be used as an activator for industrial by-products, such as ground granulated blast furnace slag in engineering applications [27–36]. The utilization of lime–slag mixture to improve the mechanical characteristics of soils is expedient for pavement engineering and some other geotechnical applications [37–41]. With an appropriate proportion of lime and slag used in the soil mixture, the early

strength of lime–slag stabilized soil tends to be much higher than that obtained with the addition of lime alone [28,32,42–45].

Pavement construction requires a great quantity of inorganic cementitious material for soil stabilization. Practically substituting slag for natural cementitious material not only alleviates the shortage of cementitious material but also helps deal with the waste of blast furnace ironmaking. However, studies related to lime–slag stabilized loess as base-course material still remain insufficient. This study related to lime–slag stabilized loess still remains insufficient. This study investigated various scenarios on lime–slag stabilized loess, by considering the effect of binder (lime+slag) content, slag content, and lime-to-slag (L/S) ratio. Laboratory standard compaction tests were performed on the mixture to obtain the optimum moisture content $\omega_{opt}$ and maximum dry density $\rho_{d,max}$. Besides, unconfined compression tests were also conducted to assess the unconfined compressive strength (UCS) at different curing periods (i.e., three, five, seven, and 28 days). Thereafter, the critical parameters slag content, L/S ratio, porosity, and porosity-to-volumetric binder content (S, L/S, η and η/L$_v$) which are governing the UCS of lime–slag stabilized loess were investigated in detail.

## 2. Experiment

### 2.1. Materials

The loess was collected from an excavation site in Lanzhou city, China. To obtain a homogeneous state in its particle distribution, the collected loess was air-dried (natural drying) and the final moisture content was around 1.56%. Then the loess was crushed down to smaller size with a rubber hammer to ensure the loess passed through the sieve with a 0.5mm aperture. Thenceforth, the physical properties of loess, such as specific gravity and Atterberg limits, were obtained by using water pycnometer method described in ASTM D854 [46] and fall cone test in accordance to the Chinese standard procedures GB/T50123 [47], respectively, and listed in Table 1. Hydrated lime with a specific gravity of 2.49 and ground granulated blast furnace slag with a specific gravity of 2.89 were purchased from Hengwang Environmental Protection Company (China). The chemical composition of loess, lime and slag was obtained by X-ray diffraction and presented in Tables 2 and 3, respectively, while physical parameters of slag provided by the manufacturer are shown in Table 4.

**Table 1.** Physical properties of the natural loess [19].

| Properties | Values |
|:---:|:---:|
| Liquid limit (%) | 26.44 |
| Plastic limit (%) | 17.31 |
| Plasticity index (%) | 9.13 |
| Specific gravity | 2.71 |
| Particle size (mm) | ≤0.5 |

**Table 2.** The chemical compositions of the natural loess.

| Compounds | $SiO_2$ | $Al_2O_3$ | CaO | MgO | $K_2O$ | $Fe_2O_3$ |
|:---:|:---:|:---:|:---:|:---:|:---:|:---:|
| Values (%) | 50 | 8~15 | 10 | 2~3 | 2 | 4~5 |

**Table 3.** Chemical compositions of lime and slag.

| Compounds | Binder (%) | |
| :---: | :---: | :---: |
| | Slag | Lime |
| CaO | 43.18 | - |
| $SiO_2$ | 31.57 | $\leq 2$ |
| $Al_2O_3$ | 15.27 | - |
| MgO | 6.68 | $\leq 2$ |
| S | 1.08 | - |
| $TiO_2$ | 0.742 | - |
| $K_2O$ | 0.448 | - |
| $Fe_2O_3$ | 0.431 | - |
| $Na_2O$ | 0.212 | - |
| $Ca(OH)_2$ | - | $\geq 94$ |
| $CaCO_3$ | - | $\leq 4$ |
| Pb | - | $\leq 3ppm$ |
| As | - | $\leq 0.4ppm$ |
| Free moisture | - | $\leq 1.0$ |

**Table 4.** Physical properties of slag.

| Properties | Measured |
| :---: | :---: |
| Specific gravity | 2.89 |
| Specific surface area($m^2$/kg) | 425 |
| Moisture content (%) | 0.28 |
| Strength grade | S95 |
| Mobility ratio (%) | 102 |
| Chloride (%) | 0.036 |

*2.2. Preparation of Specimens*

2.2.1. Compaction Tests

In this paper, standard compaction tests were carried out to determine the maximum dry densities and the optimum moisture contents of the stabilized loess on all mix ratios following procedures described in ASTM D698 [48]. The mix ratios are as shown in Table 5 with proposed binder contents (lime + slag) of 20%, 30%, and 40% and classified under the category of I (80% of loess), II (70% of loess), and III (60% of loess), respectively. According to the proportion in Table 5, the dry mixture was mixed using a cement mortar mixer. The required amount of water was then added to the dry mixture, and the fast mixing process was continued until a uniform mixture was obtained. Distilled water was then added to the mixture at an increment interval of 2% moisture content by dry weight of soil to establish its compaction characteristic between each scenario. Moisture contents thus cover the range between 14–22%, 13–21%, and 12–20%, respectively for different mix ratio under Category I; those of Category II fall in the range of 18–26%, 17–25%, and 15–23%; whereas for Category III, moisture content vary within 20–28%, 19–27%, and 17–25%. The mixture was then placed into a sealed plastic bag for 24 h under controlled temperature and humidity to achieve moisture equilibrium (note that the humidity in the standard compaction tests is not the same, and there are at least five moistures in each mix ratio (see Figure 2), while the temperature is indoor temperature and the time of compaction tests for each sample is very short). This was followed by adding the remaining predetermined amount of slag (according to Table 5) into the mixture as shown in Figure 1 (I-1, $\omega_{opt}$ **(%)**) and mixed thoroughly prior to carrying out compaction. Thereafter, the lime–slag–loess mixtures were compacted into three layers inside the compaction mold. At the final compacted layer, the surface was ensured to not exceed 6 mm of the upper edge of the mold chamber. Soil samples were obtained from the middle part of the compacted sample and dry in the oven at temperature of 105 °C to determine its moisture content.

**Table 5.** Optimum moisture content and maximum dry density for various recipes.

| Category | Mix Ratio by Mass Lime: Slag:Loess | Binder (Lime + Slag) Content (%) | Lime Content (%) | Slag Content (%) | Lime to Slag (L/S) Ratio | Slag Ratio in Binder (%) | Maximum Dry Density $\rho_{d,max}$ (g/cm³) | Average $\rho_{d,ma}$ (g/cm³) | Optimum Moisture Content $\omega_{opt}$ (%) | $\omega_{opt} - 1$ (%) | $\omega_{opt} - 2$(%) | Average $\omega_{opt}$ (%) |
|---|---|---|---|---|---|---|---|---|---|---|---|---|
| I-1 | 18:2:80 | | 18 | 2 | 9 | 10 | 1.58 | | 18.91 | 17.91 | 16.91 | |
| I-2 | 14:6:80 | 20 | 14 | 6 | 2.33 | 30 | 1.60 | 1.61 | 18.31 | 17.31 | 16.31 | 18.37 |
| I-3 | 10:10:80 | | 10 | 10 | 1 | 50 | 1.64 | | 17.88 | 16.88 | 15.88 | |
| II-1 | 27:3:70 | | 27 | 3 | 9 | 10 | 1.49 | | 22.67 | 21.67 | 20.67 | |
| II-2 | 21:9:70 | 30 | 21 | 9 | 2.33 | 30 | 1.54 | 1.54 | 21.12 | 20.12 | 19.12 | 21.00 |
| II-3 | 15:15:70 | | 15 | 15 | 1 | 50 | 1.60 | | 19.22 | 18.22 | 17.22 | |
| III-1 | 36:4:60 | | 36 | 4 | 9 | 10 | 1.41 | | 24.06 | 23.06 | 22.06 | |
| III-2 | 28:12:60 | 40 | 28 | 12 | 2.33 | 30 | 1.50 | 1.49 | 23.24 | 22.24 | 21.24 | 22.89 |
| III-3 | 20:20:60 | | 20 | 20 | 1 | 50 | 1.55 | | 21.37 | 20.37 | 19.37 | |

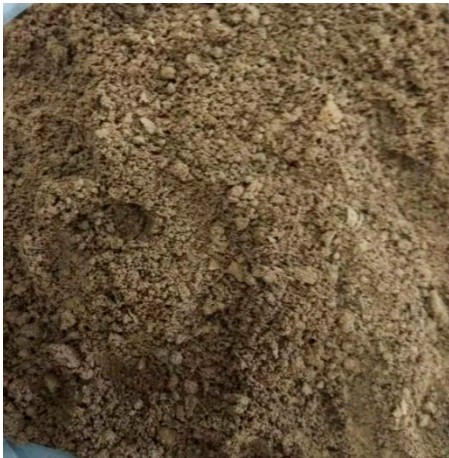

**Figure 1.** Lime–slag–loess mixture.

2.2.2. Unconfined Compression Tests

The maximum dry density and optimum moisture content obtained from the compaction tests as tabulated in Table 5 were utilized to prepare the specimens for unconfined compression tests. Three scenarios were investigated to obtain the strength properties of the mixture: (i) $\omega_{opt}$, (ii) $\omega_{opt} - 1\%$, and (iii) $\omega_{opt} - 2\%$. Note that $\omega_{opt}$ corresponds to $\rho_{d,max}$, and $\omega_{opt} - 1\%$ corresponds to $\rho_{d,max} - 0.1$, while $\omega_{opt} - 2\%$ corresponds to $\rho_{d,max} - 0.2$. The preparation process of sample for UCS test was similar to that of the compacted test soil sample. The mixture was compacted in five layers into a 125-mm-height and 61.8-mm-internal diameter cylindrical steel mold to acquire a homogeneous lime–slag stabilized loess. After the molding process, the specimens were extracted from the mold, labeled and placed in a covered container immediately to prevent further loss of moisture. After all specimens have been prepared, they were cured in a humid room at a controlled humidity of above 95% and relative temperature of $23 \pm 2$ °C for three, five, seven, and 28 days, respectively. Then, a series of unconfined compression tests were performed on the specimens according to ISO/TS 17892 [49], with displacement rate of 1.2 mm/min.

**3. Results and Discussion**

*3.1. Standard Compaction Tests*

The results on dry density versus moisture content obtained from compaction tests of lime–slag stabilized loess with various mix ratios are plotted in Figure 2. It can be seen that an increase in moisture content yields the rise of dry density through its maximum value (i.e., maximum dry density). The moisture content corresponding to the maximum dry density is defined as the optimum moisture content [3,5,10,41]. However, the dry density dropped with increasing moisture content beyond the optimum point (i.e., wet side of the compaction curve). Key parameters (i.e., optimum moisture content $\omega_{opt}$ and maximum dry density $\rho_{d,max}$) obtained from the compaction tests are listed in Table 5, which indicate that the average optimum moisture content increases with binder content, while the average maximum dry density decreases with the binder content. These results also imply that after lime and slag were added to the loess, more water was needed to achieve the optimum density as water was required to react in the hydration process. The decrease in maximum dry density is related to the flocculation of soil particles and the production of cementitious compounds [6]. When lime was added to loess with water, cation exchange of calcium quickly occurs in the lime–soil mixture, which causes flocculation of inter-particles and the decrease of loess plasticity, and thus results in extra effort required to compact the mixture [10]. For each category (I, II, and III), the maximum dry density was found to be increased with the slag ratio. This is due to the fact that slag possesses higher specific gravity than lime. Figure 3 shows the effect of lime content on the maximum dry density. For all three types of soil under investigation, it was found that the value of $\rho_{d,max}$ drops with the lime content and

increases with the slag content. The relation of the maximum dry density ($\rho_{d,max}$) and lime content ($L$) can be described by Equation (1), with a good correlation factor of $R^2 = 0.99$.

$$\rho_{d,max}(g/cm^3) = 1.727 - 0.0086 \times L \tag{1}$$

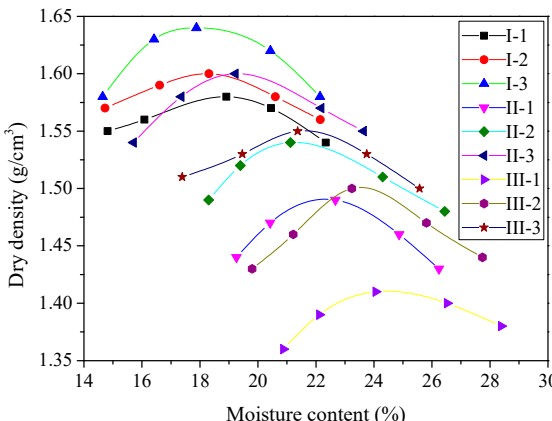

**Figure 2.** Compaction curves of lime–slag stabilized loess.

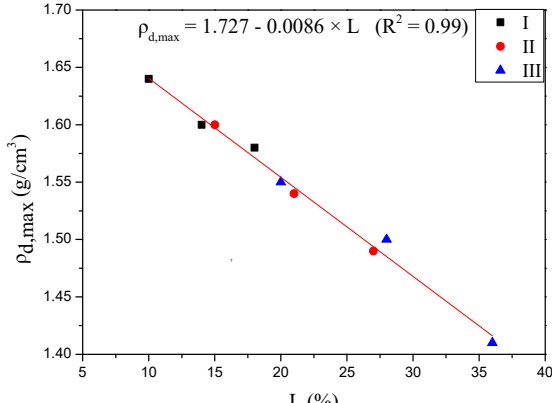

**Figure 3.** Effects of lime content on the maximum dry density $\rho_{d, max}$.

### 3.2. Unconfined Compression Tests

#### 3.2.1. Effect of the Slag Content, L/S Ratio, and Porosity

Followed by more reliable correlation established above, the specimens of stabilized loess with distinct values of density (i.e., $\rho_{d,max}$, $\rho_{d,max} - 0.1$, $\rho_{d,max} - 0.2$) and slag content were further used in unconfined compression tests, and the test results were plotted in Figure 4. It can be seen that for a certain slag content, UCS increases with the density or curing time. Linear relationships between UCS and slag content could be obtained for all curing times investigated in this study, which was in consistency with previous findings [27,31,50–52]. In addition, the early strength of lime–slag stabilized loess was significantly increased compared to previous research results [53] on lime stabilized soil. In the case of lime–soil mixture in general, the dissolved of lime provides a highly alkaline environment, in which the silicate and a small amount of aluminate ions was produced by the dissolution of soil particles. Therefore, the main cementitious material was calcium silicate hydrate, which is formed by the reaction of silicate and calcium ions in an alkaline state [54], and this process was referred to as pozzolanic reaction, in which the active components, such as silica and alumina, of loess react with calcium hydroxide of lime to form chemical products, such as calcium silicate hydrate, calcium aluminate hydrate, or calcium sulphate aluminate hydrate [6,7,26]. The introduction of

slag in the lime–soil mixture changes the reactants of the pozzolanic reaction, as well as provided additional alumina, calcia, silica, and magnesia to the lime–soil mixture [55]. Moreover, these reactants acquire high reactivity due to alkaline environment provided by the existence of enough lime for the pozzolanic reaction of slag, which was to produce more calcium silicate hydrate and calcium aluminate hydrate [26,56]. Thus the product of pozzolanic reaction was the main contribution factor to enhance the strength of the mixture [36].

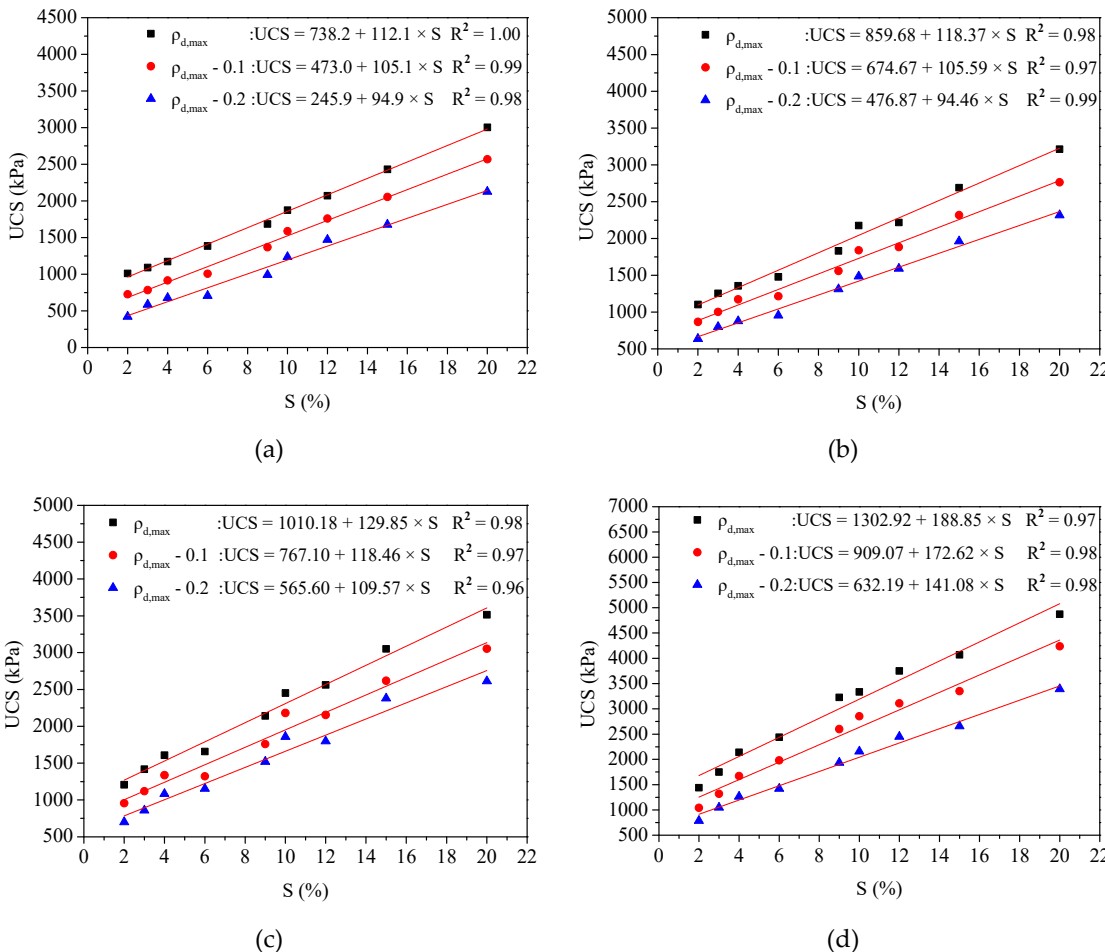

**Figure 4.** Effect of slag content on unconfined compressive strength (UCS): (**a**) 3 days curing, (**b**) 5 days curing, (**c**) 7 days curing, and (**d**) 28 days curing specimens.

Figure 5 shows the effect of L/S ratio on the UCS of stabilized loess. The results demonstrate that a drastic drop was observed in UCS with increasing L/S ratio at all curing times. The result also implies that processive substitution of lime with slag may produce significant improvements in strength development for lime–slag stabilized loess with a certain binder (lime + slag) contents, thus having notable practical applications.

Figure 6 illustrates the effect of porosity $\eta$ on UCS. It reveals that UCS decreases almost linearly with the porosity. The mechanism of the reduction in porosity influencing the soil–lime–slag strength can be explained as the variation of the amount of interlocking in soil [57–59]. It was noteworthy that porosity $\eta$ can be calculated by Equation (2) [20,28,53,60–62]. The nomenclatures for all the parameters are summarized in Nomenclature.

$$\eta = 100 - \frac{100 \left[ \frac{v_s \rho_d \left( \frac{Lo}{100} \right)}{Gs_{Lo}} + \frac{v_s \rho_d \left( \frac{S}{100} \right)}{Gs_S} + \frac{v_s \rho_d \left( \frac{L}{100} \right)}{Gs_L} \right]}{V_S} \quad (2)$$

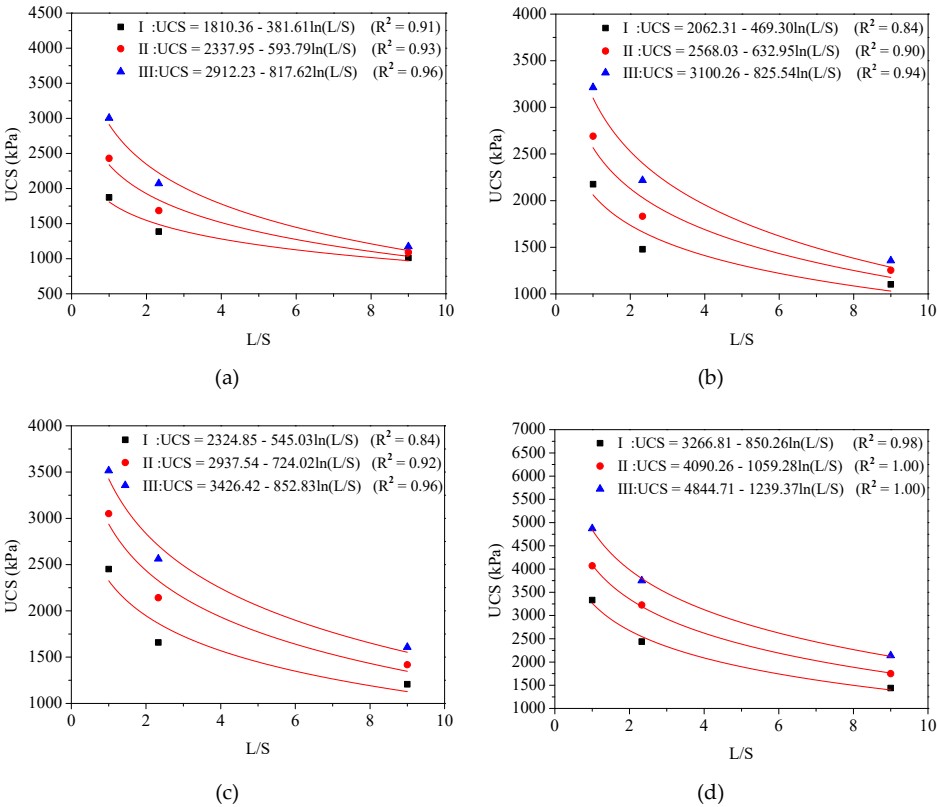

**Figure 5.** Variation of UCS with lime to slag (L/S) ratio: (**a**) 3 days curing, (**b**) 5 days curing, (**c**) 7 days curing and (**d**) 28 days curing.

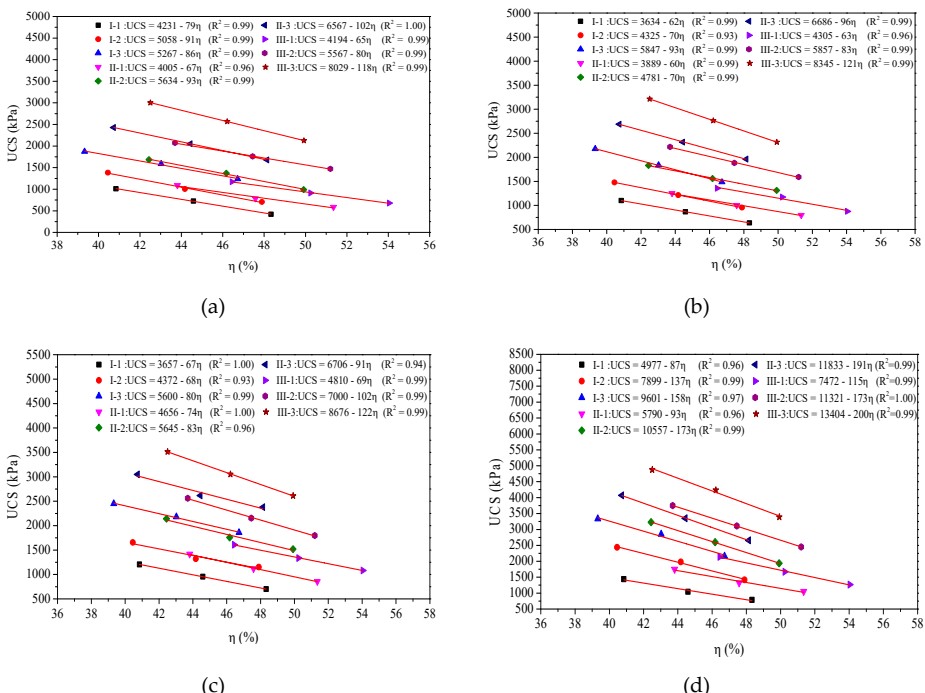

**Figure 6.** Variation of UCS with porosity: (**a**) 3 days curing, (**b**) 5 days curing, (**c**) 7 days curing, and (**d**) 28 days curing.

The specific gravities of the three materials are as follows: slag $Gs_s$ = 2.89, lime $Gs_L$ = 2.49, and loess $Gs_{Lo}$ = 2.71, whereas the specimen volume (61.8 mm diameter × 125 mm height) was $V_s$ = 375 mm$^3$. Values of $\rho_d$, $Lo$, $S$, and $L$ were adopted from Table 5.

### 3.2.2. Effect of Porosity-Volumetric Binder Content Ratio

Given the established facts that higher porosity causes the reduction of UCS, while increased binder content leads to higher UCS [50,53,63,64], attempts have been made in trying to establish a relationship between UCS and $\eta/L_v$ (for each binder content and curing time), in which $\eta/L_v$ can be characterized by Equation (3):

$$\frac{\eta}{L_v} = \frac{\text{Porostiy}}{\text{Volumetric binder content}}. \tag{3}$$

Figure 7 shows that the value of $\eta/L_v$ is the governing factor for UCS of lime–slag stabilized loess, and linear relationships can be obtained between UCS and $\eta/L_v$ for each case. A similar trend can be found for other soil types previously investigated by other researchers [65–68].

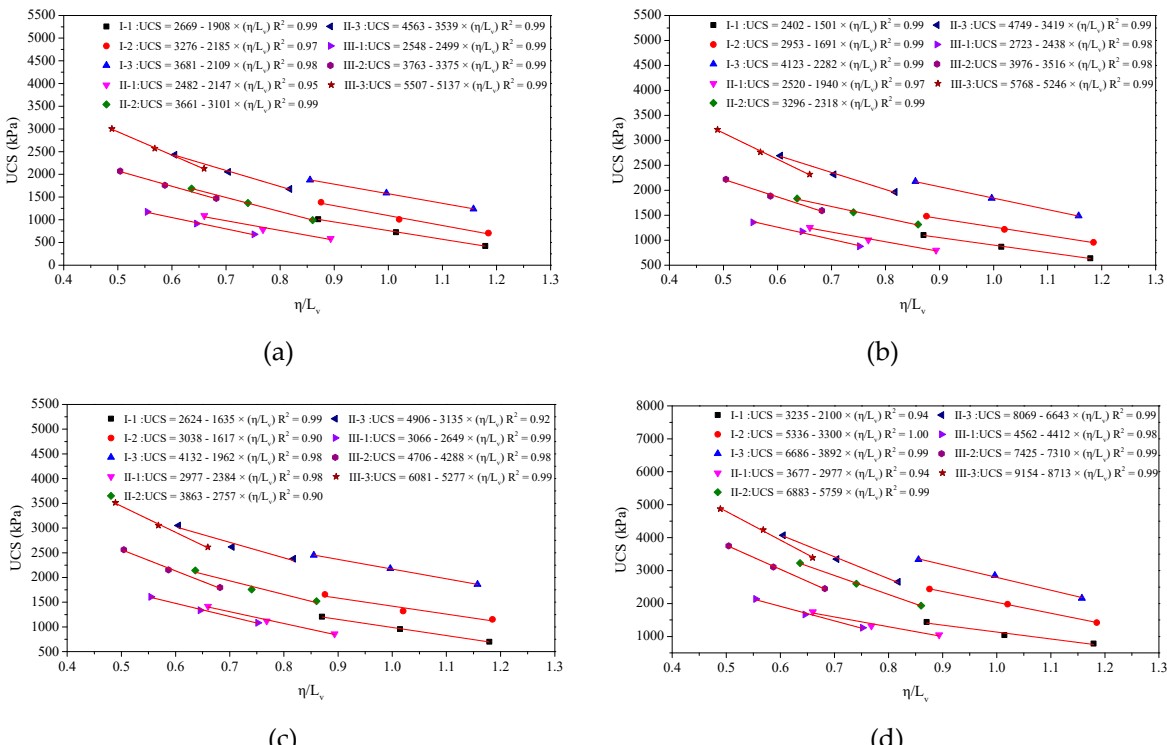

**Figure 7.** Variation of UCS with $\eta/L_v$: (**a**) 3 days curing, (**b**) 5 days curing, (**c**) 7 days curing, and (**d**) 28 days specimens.

Figure 8 illustrates a more unified relationship that can be developed between UCS and $\eta/L_v$ for the mixture with L/S ratios by applying a power of 0.34 to the parameter $L_v$. It is interesting to note that the exponential function works very well for characterizing the strength of the lime–slag stabilized loess with various curing periods.

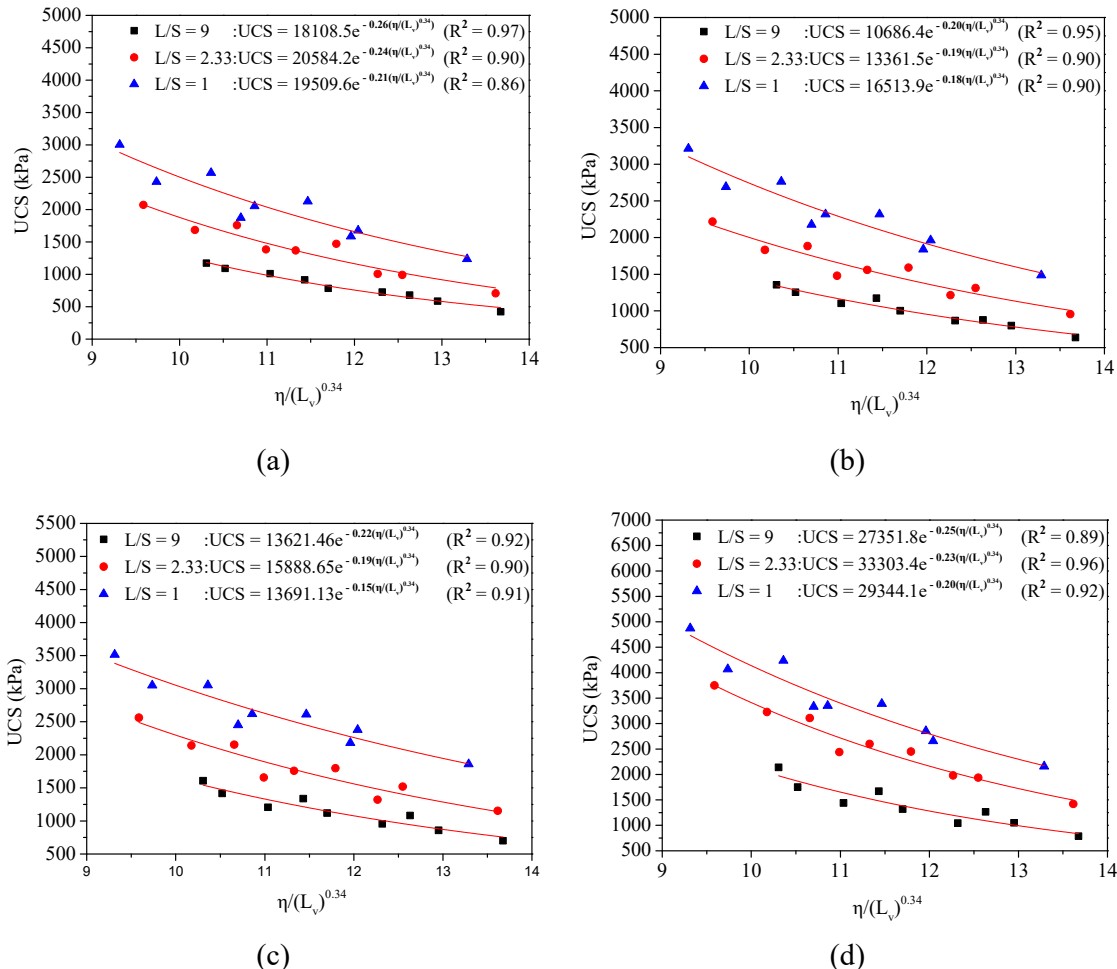

**Figure 8.** Correlation of UCS with $\eta/(L_v)^{0.34}$: (**a**) 3 days curing, (**b**) 5 days curing, (**c**) 7 days curing, and (**d**) 28 days curing.

As illustrated in Figure 8a, at three curing days, the correlation between UCS and $\eta/(L_v)^{0.34}$ with various *L/S* ratios (9, 2.33, and 1, respectively) could be represented by Equation (4)–(6), respectively:

$$\text{UCS(kPa)} = 18108.5 e^{\left(-0.26 \times \frac{\eta}{(L_v)^{0.34}}\right)} \tag{4}$$

$$\text{UCS(kPa)} = 20584.2 e^{\left(-0.24 \times \frac{\eta}{(L_v)^{0.34}}\right)}, \tag{5}$$

$$\text{UCS(kPa)} = 19509.6 e^{\left(-0.21 \times \frac{\eta}{(L_v)^{0.34}}\right)}. \tag{6}$$

For specimens cured to five days, Figure 8b exemplify that the relationships between UCS and $\eta/(L_v)^{0.34}$ can be represented through Equation (7)–(9):

$$\text{UCS(kPa)} = 10686.4 e^{\left(-0.20 \times \frac{\eta}{(L_v)^{0.34}}\right)}, \tag{7}$$

$$\text{UCS(kPa)} = 13361.5 e^{\left(-0.19 \times \frac{\eta}{(L_v)^{0.34}}\right)}, \tag{8}$$

$$\text{UCS(kPa)} = 16513.9 e^{\left(-0.18 \times \frac{\eta}{(L_v)^{0.34}}\right)}. \tag{9}$$

For the curves shown in Figure 8c, the correlations for each specimen that cured to seven days can be expressed through Equation (10)–(12) as below:

$$UCS(kPa) = 13621.5e^{\left(-0.22 \times \frac{\eta}{(L_v)^{0.34}}\right)},$$ (10)

$$UCS(kPa) = 15888.7e^{\left(-0.19 \times \frac{\eta}{(L_v)^{0.34}}\right)},$$ (11)

$$UCS(kPa) = 13691.1e^{\left(-0.15 \times \frac{\eta}{(L_v)^{0.34}}\right)}.$$ (12)

Similarly, constructed plots for 28 curing days shown in Figure 8d can be represented with the following fitting formulas:

$$UCS(kPa) = 27351.8e^{\left(-0.25 \times \frac{\eta}{(L_v)^{0.34}}\right)},$$ (13)

$$UCS(kPa) = 33303.4e^{\left(-0.23 \times \frac{\eta}{(L_v)^{0.34}}\right)},$$ (14)

$$UCS(kPa) = 29344.1e^{\left(-0.20 \times \frac{\eta}{(L_v)^{0.34}}\right)}.$$ (15)

Overall, the above formulas have evinced a higher correlation of coefficients, $R^2$ which is greater than 0.86. The form of the basic equation may be also applicable to other types of loess, but the fitting parameters may vary slightly, which may be related to the plasticity index of local loess.

## 4. Conclusions

This research evaluated the feasibility of utilizing lime–slag stabilized loess as sustainable pavement base materials, which is significant for pavement engineering from engineering, economic, and environmental perspectives. Based on the results of standard proctor compaction and unconfined compression tests of lime–slag stabilized loess, the following conclusions can be drawn:

The standard compaction test results strongly indicate that the maximum dry density of lime–slag stabilized loess drops with binder content, while the optimum moisture content increases with binder content. When the binder content used in the mixture is constant, an increase in the lime-to-slag ratio will lead to a decrease in the maximum dry density and an increase in the optimum moisture content.

The unconfined compression test results suggest a significant increase in UCS with binder content. When the binder content is constant, UCS is reduced at higher lime-to-slag ratios, which strongly suggests the dominant function of slag in strength development. As the porosity increases, the UCS of stabilized loess drops.

The relationship between void ratio, binder content, curing time, and UCS can be established using the fitting regression of experimental data.

**Author Contributions:** The authors confirm their contributions to the paper as follows: L.J., L.Z., and K.Y. proposed the idea and wrote the paper; J.G., and K.Y. revised the manuscript; B.L., S.M.L., and H.X. reviewed the results and approved the final version of the manuscript.

**Acknowledgments:** The support of the China Scholarship Council (No. 201706955065) is also acknowledged.

**Funding:** This study was funded by the National Natural Science Foundation of China (grant number 51568044, 51608407) and the first-class subjects of Lanzhou University of Technology (grant number 25-225209).

**Conflicts of Interest:** The authors declare no conflict of interest.

## Abbreviations

| | |
|---|---|
| UCS | unconfined compression strength (kPa) |
| $\rho_d$ | dry density (g/mm$^3$) |
| $\rho_{d,max}$ | maximum dry density(g/mm$^3$) |
| $\omega_{opt}$ | optimum water contents (%) |
| $L$ | lime content (%) |
| $Gs_{Lo}$ | specific gravity of loess |
| $Gs_S$ | specific gravity of slag |
| $S$ | slag content (%) |
| $R^2$ | coefficient of determination |
| $Lo$ | loess content (%) |
| $Gs_L$ | specific gravity of lime |
| $L_v$ | volumetric binder content (mm$^3$) |

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
