# Peer review of "Evaluation on Strength Properties of Lime–Slag Stabilized Loess as Pavement Base Material"

_sustainability, doi:10.3390/su11154099_

Round 1

Reviewer 1 Report

This paper evaluates the feasibility of using lime-slag stabilized loess as base-course material by assessing its unconfined compressive strength (UCS). The effects of binder content, lime-to-slag ratio, porosity and curing time on the UCS of the stabilized loess are investigated. Overall, the paper is well written and has provided new knowledge to stabilized loess. This manuscript is within the scope of Sustainability. A few concerns should be addressed:

1.       Grammar check thoroughly

2.    Introduction: why is lime-slag used in this study rather than cement? Cement has high reactivity and can hence provide high early-age strength.

3.       Section 2.2.1: Can you specify the controlled temperature and humidity?

4.       Figure 1: It is not necessary if this compaction apparatus is standard.

5.   Section 3.2.1: The authors attempted to provide some scientific explanations on the reaction mechanisms for lime-soil and lime-slag-soil. It could be important to understand the reactions among the three components. However, no detailed information about the pozzolanic reactivity of Loess is provided. What are the chemical compositions and mineral components in the Loess? Please refer to a recent study on the evaluation of pozzolanic reactivity of clay (Du & Pang, 2018, Journal of Cleaner Production, 198: 867-863).

Author Response

Please find our response as attched.

Reviewer 2 Report

The paper presents the results of experiments on the feasibility of using lime-slag stabilized loess as base-course material. The main point of research was to assess the unconfined compressive strength (UCS) of the modified loess material. The selected geotechnical parameters (optimum moisture content and maximum dry density) based on compaction tests indicate that the average optimum moisture content increases with binder content, while the average maximum dry density shows an opposite trend. The research showed that the application of lime-slag agents significantly improve the mechanical properties of the loess material in context of technical efficiency in engineering applications. The presented results may be very helpfull for civil engineering on the loess-covered areas.

Remarks:

China territory is partly covered by loess, whereas the Loess Plateau is a physiogeographical region.

Please, correct the formatting of Table 5.

Please, carefully double check the text of manuscript to avoid editorial mistakes.

Author Response

Please find our response as attched.

Reviewer 3 Report

1 Line 6 in Abstract, gain --- gains

2 Line 7 in Abstract, at constant binder content --- at same binder content

3 Line 5 in Introduction, reveal --- revealed 

4 2 Experimental --- 2 Experiment

5 Thenceforth, The physical properties ---  Thenceforth, the physical properties 

6  The chemical composition of lime and slag are obtained by X-ray --- was; please keep tense the same, check whole paper as accordingly.

7 2.2.1; 20, 30 and 40% ----- 20%, 30% and 40%

2.2.1; The figure 1 is showing the apparatus, could you provide the close up fig of lime-slag-loess mixture.

9 2.2.2 Table 5 can be improved to better show (i)wopt(ii)wopt-1%(iii)wopt-2% 

10 3.2.1 Follow by the more reliable correlation ---- Followed by more reliable correlation

11 check grammar of first paragraph in 3.2.1. 

12 3.2.2 the equation numbering. 

13 R^2 which is greater than 0.9. How about 0.86 and 0.89?

14 The study would be more convincing and significant if the real damage from hazards and severity of consequence can be added.

15 Did you do a sensitivity study on your experiments? /Did you repeat your testings to obtain an average results? 

Author Response

Please find our response as attched.

Reviewer 4 Report

This is a scientifically sound manuscript that addresses the joint effect of lime and slag in improving the mechanical strength of loess. There is little information on the use of slag to overcome the low mechanical strength of lime in the short term. Thus, this study makes an important contribution to the knowledge. I find the manuscript suitable for publication after minor revision:

1) Page 2 - line 3: the authors report that the critical parameters associated with the UCS were investigated. These parameters should be specified here.

2) Page 2 – Materials section: Loess samples were air-dried, but the drying conditions were not specified. Neither the final moisture content is presented.

3) Page 3 – Section 2.2.1 – line 6: Please provide more detail on how the lime and loess were mixed. What do the authors mean by well-mixed?

4) General comment: the formulas presented throughout the manuscript have a font size different from the text

5) Page 8 – Equation 3: this equation has two numbers

6) Page 10 – Obtained correlations: Since Loess Plateau is a very large area, the relations obtained in this study are equally valid/representative for loess samples of other zones besides Lanzhou and/or for a different lime-slag materials ? Please make it clear in the manuscript.

7) Page 10–11 – Conclusions: The conclusions must be shorter and ‘stronger’, definitely listed one by one. Precisely expressed conclusions are always important part of each paper. So, they may improve its value or make it lower.

Author Response

Please find our response as attched.
